# mRNA and Adenoviral Vector Vaccine Platforms Utilized in COVID-19 Vaccines: Technologies, Ecosystem, and Future Directions

**DOI:** 10.3390/vaccines11121737

**Published:** 2023-11-21

**Authors:** Ryo Okuyama

**Affiliations:** College of International Management, Ritsumeikan Asia Pacific University, Beppu 874-8577, Japan; ryooku@apu.ac.jp

**Keywords:** COVID-19, mRNA vaccine, adenoviral vector vaccine, innovation ecosystem

## Abstract

New technological platforms, such as mRNA and adenoviral vector vaccines, have been utilized to develop coronavirus disease 2019 (COVID-19) vaccines. These new modalities enable rapid and flexible vaccine design and cost-effective and swift manufacturing, effectively combating pandemics caused by mutating viruses. Innovation ecosystems, including universities, startups, investors, and governments are crucial for developing these cutting-edge technologies. This review summarizes the research and development trajectory of these vaccine technologies, their investments, and the support surrounding them, in addition to the technological details of each technology. In addition, this study examines the importance of an innovation ecosystem in developing novel technologies, comparing it with the case of Japan, which has lagged behind in COVID-19 vaccine development. It also explores the direction of vaccine development in the post-COVID-19 era.

## 1. Introduction

In the vaccine development against severe acute respiratory syndrome coronavirus 2 (SARS-CoV-2), vaccines using novel pharmaceutical modalities, such as mRNA and adenoviral vector vaccines, have been developed most rapidly and used globally. mRNA vaccines approved for coronavirus disease 2019 (COVID-19) were the first example of mRNA therapeutics in clinical use [1]. The adenoviral vector vaccine is a novel vaccine platform with one approved example of the Ebola virus [2]. As such, the progress in novel vaccine technology platforms for clinical application is a significant feature of COVID-19 vaccine development. These novel modality technologies enabled the success of unusually rapid vaccine development, which took only one year from the first report of infection to the emergency use approval of the vaccines [3]. This example reminds us of the power of cutting-edge pharmaceutical technology in providing solutions for public health crises.

Confirming the definition of the vaccine technology platform used in this manuscript, it is essential to clarify the distinctions. Vaccines can be produced either from the pathogen itself or from a part of the pathogen. Inactivated vaccines and live-attenuated vaccines represent the killed or weakened forms of the pathogen, respectively. Protein subunit vaccines and virus-like particles (VLPs) vaccines are types of vaccines that utilize specific components of the pathogen. These vaccines are produced through a protein expression system to generate recombinant proteins. On the other hand, mRNA and adenoviral vector vaccines represent vaccine processing technologies in which the gene expression formulates vaccines *in vivo*. This review focuses on discussing mRNA and adenoviral vector vaccines as vaccine technology platforms.

Vaccine technology platforms using mRNA and viral vectors are particularly useful during pandemics. Urgent vaccine development is highly desired to prevent further spread of the infection when the virus spreads quickly and has a high mortality rate. Vaccine development is a time-consuming process that usually takes 10–15 years. Unlike traditional inactivated vaccines, mRNA and viral vector vaccines do not require the cultivation of the pathogen, as they use mRNA or DNA encoding the antigenic proteins of the virus. These vaccines can be designed quickly by determining the inserted gene sequence once the whole-genome sequence of the targeted virus is identified. In recent years, next-generation sequencing (NGS) has advanced, allowing for the rapid identification of viral genome sequences [4]. These rapid vaccine development technologies are also useful for developing vaccines against variants of pathogens that frequently undergo mutations. mRNA vaccines can be manufactured relatively quickly due to the fact that mRNA can be duplicated in a cell-free situation by in vitro translation [5]. Thus, mRNA and viral vector vaccines offer advantages as technological platforms for pandemic vaccines that require speed. The mRNA and adenoviral vector vaccine technology, established during COVID-19, will likely become the cornerstone of future pandemic vaccine development strategies.

The technological development of mRNA and adenoviral vector vaccines has taken a long time and has been tested for various disease applications [6,7]. Usually, new modality technologies used for new drugs require long-term research and development before they can be put into practical use [8,9,10,11]. Additionally, optimal applications are often uncertain during the early stages of technology development, requiring various research and development approaches before practical implementation [12]. Furthermore, the practical application of cutting-edge technologies significantly relies on technology transfer from universities. Many novel technologies are invented in universities, and applied research and development of those technologies to translate them into products are frequently undertaken by university startups. Moderna and BioNTech, the companies that developed COVID-19 mRNA vaccines ahead of the world, are biotechnology startups founded by university researchers. In the case of vaccines, national security aspect necessitates government involvement. Thus, the presence of an ecosystem that includes startups founded on university technologies, investments in long-term applied research and development, government policies, and public-private partnerships plays a crucial role in advancing cutting-edge pharmaceutical technology development.

This review explains the new vaccine technology platforms established through COVID-19 vaccine development and discusses the detailed development process. It also discusses the technological strategy of vaccine development in the post-COVID-19 era and the ecosystem that supports it. The following section reviews COVID-19 vaccines approved by December 2022 and the technology platforms used for these vaccines. In the third section, the technological details and development history of mRNA and adenoviral vector vaccines, both newly established as vaccine technologies, are reviewed. The fourth section describes the detailed processes of vaccine technology development and the ecosystems supporting them in the cases of Moderna and BioNTech, which developed mRNA vaccines, and the University of Oxford, which developed ChAdOx1 nCoV-19, an adenoviral virus vector vaccine for COVID-19. In addition, the case of Japan, which has lagged behind in the development of COVID-19 vaccines, including mRNA vaccine, is described. Based on the above, the future technological direction of vaccine development and the significance of the ecosystem for technological development are discussed in the final section.

## 2. COVID-19 Vaccines Approved by the End of Year 2022

According to the COVID-19 Vaccine Tracker website (https://covid19.trackvaccines.org/, accessed on 12 September 2023), fifty COVID-19 vaccines had been approved in at least one country worldwide by 2 December 2022. This site double counts vaccines with the same ingredients in the cases where different companies developed the vaccine in different countries, the dose regimen of the vaccine was different, and the dosing route of the vaccine was different. After eliminating the duplicates, forty-two COVID-19 vaccines with different ingredients were identified (Table 1). Among the 42 vaccines, 8 were mRNA, 5 were viral vectors, 1 was DNA, 10 were inactivated, 17 were protein subunits, and 1 was a VLP vaccine.

Among the 42 vaccines, Spikevax and Comirnaty, both of which received emergency use authorization from the U.S. Food and Drug Administration (FDA) in December, 2020, and Vaxzevria, which received authorization from the Medicines and Healthcare Products Regulatory Agency at approximately the same time, were the earliest to be globally inoculated. Spikevax and Comirnaty are mRNA vaccines that originated from Moderna and BioNTech, respectively. These two mRNA vaccines are the first examples of mRNA therapeutics used clinically. Vaxzevria is an adenoviral vector vaccine that originated at the University of Oxford. The adenoviral vector vaccine is a new type of vaccine that was approved for the Ebola vaccine using human adenovirus type 26 in 2022 and has paved the way for clinical application [2]. Vaxzevria is a viral vector vaccine that uses ChAdOx1, an adenoviral vector modified from the chimpanzee adenovirus. The Jenner Institute at the University of Oxford has progressed in the development of ChAdOx1 as a new vaccine platform [13]. These examples indicate that newly introduced, cutting-edge pharmaceutical technologies have resolved the global crisis caused by unprecedented pandemics, reminding us of the importance of pharmaceutical innovations in solving medical and social issues.

Inactivated, protein subunit, and VLP vaccines have been used for many vaccines. Inactivated pathogens are used as antigens in inactivated vaccines. An inactivated vaccine is a traditional vaccine technology superior to a live-attenuated vaccine in terms of safety and has been applied to many vaccines. The manufacturing method for inactivated vaccines using embryonated eggs was established in the 1940s, and a new method using cell culture was developed in the 2000s [14]. The disadvantage of inactivated vaccines is that the culture of antigen pathogens requires considerable time and cost. During the 2009 Influenza A (H1N1) pandemic, vaccine supply fell short as a result of insufficient vaccine production capacity [15]. It has also been noted that the inactivation of the pathogens sometimes loses their antigenicity. Therefore, adjuvant administration is required to induce strong immunogenicity, and that inactivation by exposure to chemical or physical inactivating agents can induce irreversible changes in viral antigens, resulting in poor immunogenicity and weak cell-mediated and mucosal immune responses even with adjuvant administration.

A protein subunit vaccine is produced by creating recombinant proteins that possess viral antigens and using them as vaccines. Protein subunit vaccines were approved for hepatitis B and papillomaviruses and have been developed for various infectious diseases [16]. A protein subunit vaccine uses only an antigen protein. Therefore, it does not possess pathogenicity, and rapid scale-up is possible in manufacturing since antigen protein is produced in prokaryotic or eukaryotic cells [17]. However, the administration of antigen proteins does not induce strong immunogenicity; requiring the co-administration of adjuvants [18]. Protein subunit vaccines have been used in a limited number of countries and patients compared with other vaccine platforms among COVID-19 vaccines [16].

VLP vaccines are produced by expressing genes that encode viral structural proteins. VLP vaccines were approved for hepatitis B, human papillomavirus, and hepatitis E. VLP vaccines have an advantage over live-attenuated vaccines since they do not exhibit pathogenicity [19]. Furthermore, in the case of inactivated vaccines, there may be instances in which structural proteins are modified during the inactivation process. However, VLP vaccines can more closely mimic the structure of authentic vaccines, allowing for the maintenance of higher immunogenicity [19]. In addition, stronger immune responses can be expected since VLP vaccines induce immunity via the same mechanism as natural viruses due to their size, surface geometry and ability to induce both innate and adaptive immune responses [20]. At least six VLP vaccines have been developed for COVID-19 [21], and Medicago’s vaccine was approved (Table 1). Medicago possesses a unique vaccine production technology using tobacco plants, which enables easy manipulation and infiltration procedures and high expression efficiency for introduced genes [22].

Conventional vaccines have several disadvantages. The immunogenicity induced by inactivated vaccines is moderate. Therefore, a combination of adjuvant treatment and booster administration is required [23]. Pathogen inactivation is a time-consuming and costly process, making urgent vaccine development and distribution difficult [24]. Due to their low immunogenicity, the protein subunit vaccines require co-treatment with a suitable adjuvant [18]. The downstream processing is technically difficult, and a high production cost is required for the VLPs vaccine [25]. These challenges hinder the rapid development of potent vaccines to swiftly prevent the worldwide propagation of COVID-19.

mRNA and viral vector vaccines were suitable technological platforms for overcoming these challenges and rapidly developing effective COVID-19 vaccines. The first reason is that mRNA vaccines can be manufactured faster and at a lower cost than other vaccines [26]. The adenovirus vector vaccine also requires a relatively short manufacturing time, and Vaxzevria has been successfully produced at a low cost, making it suitable for meeting global needs [27]. The second reason is that a strong immune response is expected without adjuvant administration. The protein subunit vaccine requires the co-administration of an adjuvant to induce a strong immune response, as only specific antigen proteins of interest are administered. mRNA and viral vector vaccines are similar to protein subunit vaccines in that only specific antigen proteins of interest are presented within the body. However, lipid nanoparticles (LNPs), a component of the delivery vehicle, show an adjuvant-like effect. Therefore, mRNA vaccines can induce a strong immune response without adjuvant co-administration [28]. Adenoviral vector vaccines can induce a strong immune response by exhibiting an adjuvant effect through the adenovirus vector, which triggers immune reactions similar to those observed during viral infection [29]. Third, mRNA and viral vector vaccines can be designed rapidly once the viral genome sequence is identified. These vaccines can be flexibly designed for various pathogens by altering the sequences of the carried mRNA or DNA [30,31]. For the case of the COVID19 mRNA vaccine, the genetic sequence of SARS-CoV-2 was disclosed by the Chinese authorities on 11 January, 2020. On 13 January, only two days after the disclosure, Moderna announced that they had finalized the sequence for the SARS-CoV-2 vaccine [32]. This speed of the design process implies that vaccines can be rapidly developed in response to the emergence of new variants. In the case of COVID-19, the Omicron variant was first detected in late 2021 and became dominant in mid-2022. Moderna and BioNTech swiftly developed variant-adapted vaccines using their respective mRNA vaccine platforms, which were subsequently approved [33].

Including non-vaccine medications, no mRNA therapeutics were approved before the COVID-19 vaccines. Approved adenoviral vector vaccines were limited before COVID-19. Therefore, there should have been concerns regarding the urgent authorization of Spikevax, Comirnaty, and Vaxzevria. Such a rapid approval may not have been possible under normal circumstances. However, in the face of the unprecedented emergence of the COVID-19 pandemic, these three vaccines obtained emergency use authorization within a remarkably short period of just one year from the onset of the outbreak. Their effectiveness and safety have been proven in clinical settings, saving countless lives from infections. The mRNA vaccine technology has also enabled the rapid development of vaccines against mutant strains. Moderna and BioNTech, utilizing the mRNA vaccine platform, have successfully developed and obtained approval for a combination vaccine targeting the COVID-19 Omicron variant by December 2022 (Table 1). mRNA and viral vector vaccines can be seen as a new trend in post-COVID-19 vaccine development. As shown in Table 1, many inactivated and protein subunit vaccines have also been developed and used in clinical settings. In the future, a new standard vaccine development strategy will likely emerge, where various old and new vaccine platforms will be utilized to comprehensively combat emerging infectious diseases. The development of cutting-edge pharmaceutical technologies, such as mRNA therapeutics and viral vector vaccines, does not occur overnight. New innovative technologies often originate in public research institutions, such as universities. These technologies are then applied, developed, and eventually translated into practical use through collaborations between academia and industry or by transferring them to startup companies. The initial discovery leading to the idea of mRNA therapeutics was made in 1989 [34], and it took approximately 30 years until the first practical use of mRNA in COVID-19 vaccines. BioNTech was founded in 2008 and Moderna in 2010, which means that it took approximately a decade from its inception to the practical application of mRNA vaccines. The Jenner Institute initiated research on ChAdOx1 in the early 2000s [13], with nearly 20 years spent on the clinical application of viral vectors. Thus, translating cutting-edge pharmaceutical technologies into practical applications takes a long time and substantial research and development investments. Without the necessary resources for talent, funding, and infrastructure to support long-term and large-scale research and development efforts, the successful implementation of innovative pharmaceutical technologies is unlikely. In the case of COVID-19 vaccines, the long-term accumulation of technological developments under normal circumstances enabled the rapid use of innovative technologies during the pandemic. The existence of an ecosystem that made this possible has played a crucial role in the development of innovative technology.

However, Japan has failed to rapidly develop innovative pharmaceutical technologies and has lagged behind in COVID-19 vaccine development. In Japan, the startup ecosystem that bridges universities’ innovative technology seeds into applications has not matured, and the total amount of startup investment is about 1/100th that in the U.S. [35]. Consequently, in drug discovery, in which startups, including those originating from universities, play a significant role, Japan lacks international competitiveness [36]. In the case of COVID-19 vaccines, Japan failed to develop domestically produced vaccines and instead imported or clinically developed the vaccines that originated from non-Japanese companies for its citizens. Consequently, the trade deficit in pharmaceuticals expanded, exposing the weakness of Japan’s drug discovery capabilities [37]. In Japan, a pharmaceutical company began the research and development of mRNA vaccines shortly after the onset of COVID-19. However, owing to a lack of technological accumulation, they could not develop a vaccine quickly and meet the demand for vaccine administration (discussed later). Revealing the reality of COVID-19 vaccine development in Japan and comparing it to the situation in Europe and the United States serves as valuable information for contemplating the importance of the startup ecosystem in translating innovative pharmaceutical technologies into practical applications.

The following chapters focus on COVID-19 vaccine development at Moderna, BioNTech and the University of Oxford. The mRNA and adenoviral vector vaccine technologies are explained, along with a detailed account of the development process, key players, and funding sources that supported technology development. The developmental history and current status of mRNA vaccines in Japan are also discussed. Based on these insights, the direction of vaccine development in the post-COVID-19 era and the significance of the ecosystem in supporting the development of innovative pharmaceutical technologies are discussed.

## 3. New Vaccine Technology Platforms Established through COVID-19 Vaccines

### 3.1. mRNA Vaccine

mRNA therapeutics involves encapsulating exogenous mRNA within LNPs and introducing it into living organism to express a desired protein, thereby achieving therapeutic effects. This technology is also used in vaccines in which mRNA encoding the antigen protein of interest is administered to the body. This leads to the expression of the antigen protein within the body, ultimately imparting immunity to the host, thereby exerting the effect of the vaccine. The mRNA vaccine technology has been applied to COVID-19 vaccines. Two COVID-19 mRNA vaccines, originating from Moderna and BioNTech, received emergency use authorization from the FDA just one year after the onset of COVID-19. These became the first two COVID-19 vaccines widely administered globally [38]. No mRNA therapeutics were approved for commercial use before the approval of these two [6].

mRNA vaccines are safe since the mRNA encoding the antigen protein is relatively short-lived, and the risk of integrating exogenous genes into the genome is extremely low since the mRNA would not enter the nucleus [16]. The proteins expressed in the introduced mRNA induce the production of neutralizing antibodies and prevent infection. In COVID-19 mRNA vaccines, anti-spike IgG levels are associated with protection against infection [39]. The activation of innate immune pathways also contributes to increased adaptive immunity in mRNA vaccine responses. In mRNA vaccines, it is believed that in addition to the mRNA itself, LNPs activate innate immunity and function as adjuvants, resulting in robust immunostimulatory activity [28]. mRNA vaccines offer advantages in terms of manufacturing. It has a high level of safety in the manufacturing process since it does not require large-scale culture of highly pathogenic organisms and it mitigates the risk of contamination with live infectious reagents [40]. In addition, speedy manufacturing is possible, and the manufacturing cost is relatively low [26].

The concept of mRNA application in medicine has been used for approximately 30 years. The encapsulation of mRNA in cationic lipids to introduce it into cells and express proteins was first reported in 1989 [34]. Simultaneously, Malone, a researcher who reported this discovery, noted that treating RNA as a drug might be possible if cells could create proteins from the mRNA delivered [6]. In 1990, it was reported that mRNA could be administered *in vivo* and used to express the luciferase protein in mouse muscle tissue, suggesting the potential application of mRNA as a drug by introducing it into living organisms and expressing proteins [41]. However, mRNAs introduced from outside the body are unstable, the duration of protein expression is short, and the amount of expression is insufficient. Owing to these limitations, the practical application of mRNA therapeutics has not progressed significantly for a long time.

Several technological breakthroughs have enabled the clinical application of mRNA therapeutics. The first breakthrough was the control of immunogenicity. mRNA activates pattern recognition receptors, such as Toll-like receptors 7 and 8, and retinoic acid-inducible gene I [42,43]. This activation induces an innate immune response and causes fever. It was also a problem that administrated mRNAs were quickly broken down, resulting in insufficient protein expression. Kariko et al. found that the substitution of a modified uridine, called pseudo-uridine, for the natural uridine residue of mRNA reduces the immunogenicity of the mRNA [44]. Furthermore, mRNA with uridine replaced with pseudo-uridine has been demonstrated to have a higher translation efficiency into proteins than conventional mRNA [45]. However, the intrinsic immunostimulatory activity of mRNA can help induce productive immunity. Moreover, the innate immune activation ability of mRNA vaccines can vary depending on their combination with LNP composition. The optimization of modified nucleotides has progressed, and N1-methyl pseudo-uridine has been utilized in several mRNA vaccines, including Spikevax and Comirnaty [46,47].

The second technological breakthrough was progress in the optimization of mRNA structures. The mRNA used in mRNA therapeutics is produced from template DNA via *in vitro* transcription. To translate mRNA into protein within cells, it is necessary to add a cap structure at the 5′ end. However, the capping direction cannot be traditionally controlled. In 2001, the anti-reverse cap analogs method was developed, allowing for the efficient attachment of the cap structure at the 5’ end. This significantly increased mRNA synthesis efficiency [48]. Subsequently, further optimization of cap analogs occurred, and in the case of COVID-19 vaccines, analogs with a Cap1 structure where the 2’ hydroxyl group of the 5’ cap is methylated have been utilized. This improved the capping efficiency and translational properties [49]. In addition, optimization of the poly(A) tail and 3’ and 5’ untranslated regions has been pursued to enhance mRNA stability and translation efficiency [50]. The codon composition of the open reading frame is important for mRNA translation efficiency. GC-rich sequences have been shown to increase translation efficiency 100-fold compared with less GC-rich sequences [51].

The third technological breakthrough was the progress in mRNA purification methods. Double-stranded RNA (dsRNA) generated as a byproduct of mRNA synthesis induces type I interferon, reducing mRNA vaccine efficacy [52]. Therefore, dsRNA must be removed during purification [53]. Methods such as high-performance liquid chromatography [54] and cellulose adsorption [55] have been developed.

The fourth technological breakthrough was the progress in delivery technology. mRNA is unstable in the human body and it cannot penetrate the cell membrane. Therefore, mRNA must be incorporated into the LNPs for their introduction into the body. LNPs are carrier molecules made primarily from lipids, essential components of cell membranes, and are formulated with various functional molecules. Neutral nanoparticles are used in LNPs since electrostatic interactions with mRNA inhibit mRNA translation within the cytoplasm [56]. In contrast, LNPs must be positively charged in the acidic environments within the endosome. They interact with the negatively charged endosomal membrane, disrupting the endosome and facilitating the uptake of mRNA into the cytoplasm. Tertiary amines positively charged pH-dependently have been identified as LNP components [57]. Lipids, including tertiary amines, have been used by Spikevax and Comirnaty [58].

Through the various technological improvements described above, Spikevax and Comirnaty were commercialized as mRNA vaccines. Both vaccines used the same mRNA sequences. The full-length sequence of the spike protein of SARS-CoV-2 was used with two amino acid substitutions from the wild-type sequence to stabilize the expressed protein structure [59]. Spikevax and Comirnaty used nucleotides modified by substituting N1-methylpseudouridine for uridine to reduce the innate immune response of unmodified mRNA [60]. In a phase 3 study of Spikevax, the vaccine efficacy in the prevention of COVID-19 with onset at least 14 days after the second injection was 94.1%, with rare serious adverse events [61]. In a phase 3 study of Comirnaty, the vaccine efficacy in the prevention of COVID-19 with onset at least seven days after the second injection was 95% with a low incidence of serious adverse events [62]. These results have led to the clinical demonstration of the high efficacy and safety of mRNA vaccines against COVID-19.

### 3.2. Viral Vector Vaccine

Live-attenuated or inactivated pathogens have been generally used as vaccines. The main mechanism of immune response induction for live-attenuated or inactivated vaccines is their structural proteins. Although the innate immune system can sense virus-derived DNA and RNA through pattern-recognition receptors (PRPs), it is not likely that these vaccines are sensed by PRPs [63]. SARS-CoV-2 contains four major structural proteins (spike, membrane, envelop and nucleocapsid) and these proteins could mainly serve as targets of vaccine-induced immune responses [64]. However, pathogens must be cultured and proliferated in live-attenuated and inactivated vaccines, which requires considerable time and cost. Therefore, those vaccines cannot be developed rapidly when urgent vaccine development is desired during pandemics such as the COVID-19 outbreak [65].

In the case of vaccines using viral vectors as carriers, the viral vector mimics the immune responses induced by natural viral infection. Therefore, strong vaccine efficacy is expected [66]. Furthermore, the pathogenesis of the virus itself is not considered the gene encoding the antigen of the pathogens of interest that is delivered by the viral vector and expressed in the body [67]. The sequence encoding an antigen can be easily designed once the genome sequence of the pathogen is identified, and various antigen sequences can be tested [67]. Currently, a whole genome sequencing of a virus is possible within a short time, owing to progress in next-generation sequencers [68]. For these reasons, viral vector vaccines have been highly anticipated as a vaccine technology that induces a strong immune response, is safe, and enables rapid and simultaneous vaccine development. Many viral vectors derived from adenovirus, vaccinia virus, measles virus, and vesicular stomatitis virus have been tested as vaccines [40]. Viral vector vaccines using vesicular stomatitis virus were developed and commercialized and contributed to the prevention of spread of the Ebola virus outbreak in 2014 [69].

Among the viral vectors, many technological developments have been made in adenoviral vectors in recent years. Adenoviruses are double-stranded DNA viruses with a genome of approximately 34–43 kb, amenable to easy manipulation [70]. With the exception of vaccines, drugs using adenoviral vectors have been studied and developed for various diseases, including cancer, cardiovascular diseases, metabolic diseases, neurological diseases, muscular diseases, and immune deficiency [71]. Several drugs have been approved for use in the field of oncology. Gendicine is a gene therapy that delivers p53, a tumor suppressor gene, to cancer cells using adenoviral vectors to arrest their cell cycle. Gendicine was approved as the first commercial gene therapy product for head and neck squamous cell carcinoma by the Chinese State Food and Drug Administration (SFDA) in 2003 [72]. Oncorine is an oncolytic virus transmitted to cancer cells to induce cell death. This drug uses a gene-manipulated adenovirus called H101 and was approved for nasopharyngeal carcinoma by the SFDA in 2005 [73].

The adenoviral vector has been successfully used clinically, and its safety in humans has been secured. Therefore, it has been used as a technological platform for vaccines [70]. Chimpanzee’s adenovirus and human adenovirus type-5 and type-26 have been mainly used as vaccines. ChAdOx1, a chimpanzee’s adenovirus developed by the University of Oxford, was used for Vaxzevria. The issue with using human-derived adenoviruses as vectors is that humans may already have neutralizing antibodies against the adenovirus, which raises concerns about a weakened clinical effect [74]. To avoid this issue, ChAdOx1 was established based on simian adenovirus type-Y25, a neutralizing antibody against which exists in 0% of UK adult sera and 9% of Gambian adult sera [75]. Vaxzevria is a viral vector vaccine that incorporates the full-length sequence of the spike protein of SARS-CoV-2 into the ChAdOx1 [76]. In a phase 3 study of Vaxzevria, the vaccine efficacy in the prevention of COVID-19 with onset 15 days or more after the second injection was 74% [77].

Adenovirus type 5 (Ad5) is one of the most common and well-characterized human adenoviruses [78]. Viral vector vaccines based on Ad5 have been developed for various pathogens, including Ebola [79] and *Trypanosoma cruzi* [80]. However, pre-existing anti-vector immunity can attenuate the immunogenicity of the Ad5 vector vaccine [81]. In the development of an Ad5 vector vaccine for HIV, clinical trials were halted, particularly in the group of subjects who had pre-existing antibodies against Ad5, as an increase in HIV infection rates was observed compared with the placebo group [82]. For COVID-19, CanSino Biologics and the Beijing Institute of Biotechnology developed a vaccine using an Ad5-based vector with E1/E3 deletions to eliminate replicability, carrying the gene encoding the spike protein of SARS-CoV-2 [83]. In contrast to HIV, this vaccine demonstrates sufficient efficacy against SARS-CoV-2 infection. In a phase 3 study, one dose of this vaccine showed 57.5% efficacy against symptomatic COVID-19 infection [84]. It has been reported that the vaccine’s efficacy decreases in individuals with strongly positive pre-existing antibodies against Ad5 [85].

To overcome pre-existing immunity issue of Ad5, vector development using adenovirus serotypes with lower prevalence in humans has also been pursued. One of them is Ad26 [86]. Human Ad26 has a low prevalence of pre-existing antibodies, and antibody titers are low even if present. Therefore, Ad26 vectors have been widely used for vaccine development. Vaccines against Ebola hemorrhagic fever have been approved in the EU, and clinical development is underway for HIV, malaria, RS virus, Filo virus, Zika virus, and human papillomavirus using Ad26 vectors [87]. For COVID-19, Janssen developed Jcovden, a viral vector vaccine that utilizes a modified Ad26 vector with E1/E3 deletions and incorporates a sequence of the SARS-CoV-2 spike protein with amino acid substitutions for stabilization [88]. This vaccine protects against moderate to severe-critical COVID-19 with an onset of at least 14 days after administration, with 66.9% efficacy [89]. Gamaleya developed gam-COVID-vac, a viral vector vaccine for COVID-19. This vaccine consists of Ad26-based vector for the prime dose and an Ad5-based vector for the boost dose [90].

## 4. Research and Developmental History of mRNA and Adenoviral Vector Vaccines

Various players and their interactions are required to develop and commercialize cutting-edge technologies. The national innovation system of each country is formed by the main players in creating innovation, such as corporations, universities, governments, and their relationships [91]. This system creates uniqueness in each country’s economic growth and industrial competitiveness [91]. Each player mutually compensates rather than independently functions. For instance, universities’ innovative technological seeds are not linked to commercialization without human resources and companies that bridge the seeds into applications and investments that support applied research and development. To realize commercialization, university scientists, entrepreneurs, investors, and incumbent large companies should build ecosystems with mutual relationships and facilitate innovation. The role of the government is also important since the establishment of university technology transfer policies, public supports for research and development, and venture promotion measures strongly affect the enhancement of innovation.

Figure 1 shows the ecosystem framework in cutting-edge technology development towards commercialization. The role of public research institutions in generating technological seeds from their basic research is crucial for innovations using cutting-edge technologies. University startups often conduct applied research and development to bridge technological seeds for commercialization. Entrepreneurial teams, including entrepreneurial scientists with strong technology expertise and corporate managers with business expertise, are required to establish startups. In the early stages of startups, when the risk of commercialization is still huge, public funds and angels financially support the achievement of proof of concept. Once the proof of concept is confirmed and application development is on the horizon, investments from venture capitalists and collaboration with large companies interested in technology support large investments towards commercialization. Policy development and public support for funding and human resources sometimes play important roles in facilitating these processes. The development of mRNA vaccines and the ChAdOx1 viral vector vaccine discussed in this article has been supported by various players. In the following sections, an overview of the history and contributions of the players in each case is provided.

### 4.1. Moderna

Kariko and Weissman, who received Nobel Prize in Physiology or Medicine in 2023, discovered in 2005 that the replacement of the uridine residue of mRNA with pseudo-uridine reduced side effects, such as the activation of innate immune system while maintaining the translational properties of mRNA into proteins [44]. Rossi, who worked as a stem cell biologist at Boston Children’s Hospital at that time, was inspired by the induced pluripotent stem (iPS) cell development technology that Yamanaka discovered and was working on research to create safe iPS cells that introduced genes that were not integrated into genome [92]. Rossi et al. transformed skin cells into embryonic-like stem cells by gene transfection using modified mRNA and differentiated them into muscle cells [93]. This research garnered significant attention at the time, and Rossi was selected as the Time Magazine’s Person of the Year in 2010 [94]. Rossi felt that this technology had significant potential for human therapeutics, which motivated him to establish Moderna. At that time, he considered its broad applicability across approximately 6000 genetic diseases rather than focusing on a specific disease, and vaccines were not high on the priority list [92]. This research caught the attention of Afeyan, CEO of the Cambridge biotech investment firm Flagship Pioneering [95]. Afeyan is the founder of Flagship Pioneering and has cofounded and helped build over 70 life science and technology startups during his career [96]. Additionally, a professor of chemical engineering at MIT and a renowned serial entrepreneur, Langer, took an interest in this technology and became the co-founder of Moderna [97]. Both were attracted to the wide range of possibilities that modified mRNA technology could offer [95].

Moderna’s technology has attracted the interest of major pharmaceutical companies. In 2013, AstraZeneca entered into an exclusive agreement with Moderna to research, develop, and commercialize treatments in cardiovascular, metabolic, and renal diseases and cancer. The contract included an upfront payment of $240 million and subsequent milestone fees totaling $180 million [98]. In 2014, Alexion entered into an exclusive agreement with Moderna for mRNA drug development in rare diseases, which included a $100 million payment and a $25 million investment [99]. Merck has been engaged in joint research and development with Moderna in personalized cancer medicine since 2016. In the same year, they paid $200 million [100], and in 2018, they invested an additional $125 million in Moderna [101]. Moderna secured substantial funding from the market. In 2013, Moderna raised $450 million in a financial round, setting a record for the highest amount ever raised by a privately held biotech company [102].

Moderna has received investment and human resource support from the government. In 2013, the Defense Advanced Research Projects Agency of the United States Department of Defense (DARPA) awarded Moderna up to $25 million for the research and development against Chikungunya infection [103]. In 2016, the Biomedical Advanced Research and Development Authority of the U.S. Department of Health and Human Services awarded Moderna up to $125 million to fund their Zika vaccine program [103]. The National Institute of Allergy and Infectious Diseases (NIAID) and Moderna have conducted a four-year collaboration on HIV and emerging infectious diseases, and three NIAID scientists joined this collaboration [104]. Regarding the COVID-19 vaccine, NIAID announced that this vaccine had been co-developed with scientists at NIAID and Moderna [105]. Moderna has paid $400 million to the government for a chemical technique key to its vaccine. However, there is a patent dispute between the two parties over a different vaccine patent [106].

Moderna had no approved drug product in the market before the approval of the COVID-19 vaccine. Their research and development efforts over approximately ten years, from their inception to the first drug launch, were made possible by substantial funding and research support from investors, large corporations, and the government. Moderna began patent applications shortly after its founding in 2010 and had filed 90 patent applications by the end of 2019 when COVID-19 emerged [12]. They had published 44 research papers by the end of 2019, including 18 papers on foundational technology for mRNA therapeutics, twelve papers on infectious diseases (excluding COVID-19), seven papers on rare diseases, and two papers on tumors [12]. The papers related to foundational technology for mRNA therapeutics covered a wide range of techniques, including those related to mRNA structure, such as open reading frame and 5’ UTR sequence optimization, delivery technology, such as the composition and size of LNPs, methods for mRNA purification and synthesis, temperature stability, and other aspects pertaining to the production of mRNA therapeutics [12]. Moderna possesses proprietary digital tools that enable rapid mRNA design and a highly automated production facility [107]. Before the development of the COVID-19 vaccine, Moderna established a diverse range of technologies and facilities related to the design, synthesis, and manufacture of mRNA therapeutics by leveraging the substantial funds acquired.

Before developing the COVID-19 vaccine, Moderna conducted clinical trials on several other diseases and gained experience in the clinical application of mRNA therapeutics. Starting with the influenza vaccine trials in 2015, they had conducted nine clinical trials for vaccines against infectious diseases (excluding COVID-19) and four clinical trials for cancer vaccines by the end of 2019. Additionally, they conducted two clinical trials to treat rare diseases [12]. As of 2021, mRNA vaccine trials against infections other than COVID-19 have been conducted using 18 compounds, 13 Moderna compounds [108]. From these accomplishments, it is evident that Moderna had already acquired substantial knowledge and experience in the clinical application of mRNA therapeutics for vaccines and other diseases before the emergence of COVID-19.

### 4.2. BioNTech

BioNTech is a biotechnology startup founded in 2008 by the husband-and-wife team of Şahin and Türeci. Şahin is involved in cancer immunotherapy research, while Türeci is a physician and immunologist. The two have led a research team at the University of Mainz since 2000 [109]. Since then, Şahin and Türeci have developed several cancer immunotherapy platforms. Initially, they researched antibodies that activate immune effectors to attack tumors. To develop these drugs, they founded Ganymed in 2001 [109]. Ganymed developed a therapeutic antibody called IMAB362 that targets claudin 18.2, a protein highly expressed in pancreatic neoplasms [110]. IMAB362 combined with first-line chemotherapy exhibits a clinically relevant benefit in progression-free survival and overall survival and a favorable risk/benefit profile [111]. This result gained the attention of pharmaceutical companies, and Astellas acquired Ganymed for $1.4 billion in 2016 [112].

In the late 1990s, Gilboa et al. introduced mRNA encoding cancer antigens into dendritic cells at the Duke University Medical Center. This leads to antigen presentation within the body, activates the immune system, and demonstrates the potential for attacking cancer [113]. Gilboa et al. established Merix Bioscience and conducted clinical trials of cancer vaccines using this technology [114]. Although the large-scale clinical trials for this vaccine candidate failed several years later, Şahin was aware of this research and decided to pursue a similar approach by directly administering mRNA into the body [6]. Türeci and Şahin believed that mRNA vaccine technology had matured as a platform to advance personalized cancer vaccines. Consequently, BioNTech was founded in 2008 [109]. Since its inception, BioNTech has been supported by substantial investments. Thomas Struengmann, a prominent German investor, met with Şahin and Türeci at the founding of BioNTech and was moved by the potential of their technology and their passion, leading him to make significant investments [115]. In 2008, BioNTech secured $180 million in funding as part of its seed round [116]. In 2018, BioNTech raised $270 million in a Series A financing round led by the Redmile Group, with participation from multiple venture capital firms [116]. Before gaining approval for the COVID-19 vaccine, BioNTech did not have any previously approved drugs, and substantial research and development investments had been primarily covered by external sources [12]. Development of the COVID-19 vaccine involved a joint effort between BioNTech and Pfizer. However, this was not their first collaboration. In 2018, they entered into a collaborative agreement with $305 million to develop an mRNA vaccine against influenza [117]. Owing to this pre-existing relationship, BioNTech approached Pfizer during the development of the COVID-19 vaccine [118].

With this abundant support, BioNTech has been conducting research and development for over a decade, from its inception to the approval of the COVID-19 vaccine. Şahin and Türeci’s group has been publishing research papers on the fundamental technology of mRNA therapeutics since the mid-2000s, even before the establishment of BioNTech, and papers under the name of BioNTech have been consistently published since 2013 [12]. These studies cover a wide range of technologies essential for mRNA therapeutics, including those related to poly(A) tail and 3′ UTR sequences that enhance mRNA stability and translational efficiency, structural optimization of the 5′ cap region, and optimization of LNP delivery systems [12]. BioNTech has also researched to enhance the manufacturing efficiency of mRNA therapeutics. By making minor adaptations to the manufacturing technology established before the onset of COVID-19, they enabled the scale-up of COVID-19 vaccine production to more than one billion doses [119].

BioNTech began filing patents under the name BioNTech as early as 2002, before its establishment, and by 2020, had filed 165 patent applications registered in Espacenet [12]. Many patents related to the optimization of the mRNA structure and delivery technology have been filed since the 2000s. Recently, numerous patents have been filed concerning the formulation and storage of mRNA vaccines [12]. Several clinical trials have been conducted since 2012. By the end of 2019, 13 clinical trials had been initiated, and all clinical trials conducted before the development of the COVID-19 vaccine were related to cancer vaccines [12]. Studies on mRNA vaccines targeting viruses other than SARS-CoV-2 have been conducted. In 2017, the effectiveness of a vaccine against the Zika virus was demonstrated [120]. BioNTech’s strength is evident in its talent acquisition capacity. As mentioned previously, the practical application of mRNA therapeutics requires a reduction in mRNA immunogenicity. The discovery that changing uridine to pseudo-uridine, as demonstrated by Kariko et al., can lower immunogenicity while maintaining translational efficiency was a significant technological breakthrough [44]. Kariko began her research on mRNA therapeutics in 1989 and, alongside her colleague Weissman, has been a pioneer leading the world in the technological development of mRNA therapeutics [121]. Based on the discovery of pseudo-uridine, Kariko et al. founded a startup named RNARx to develop mRNA therapeutics. However, due to a lack of funds, the company ceased operations in 2013 [6]. In 2013, Kariko delivered a guest lecture at University Medical Center Mainz and met Şahin. Kariko was recruited to join BioNTech and subsequently made significant contributions to the research and development of the COVID-19 vaccine as Senior Vice President at BioNTech [122].

### 4.3. University of Oxford

ChAdOx1, an adenoviral vector used in Vaxzevria, is a vaccine platform developed by the Jenner Institute at the University of Oxford. In the early 2000s, the Jenner Institute developed a malaria vaccine using adenoviruses. Promising results were achieved using a chimpanzee-derived adenovirus in collaboration with the Italian company Okairos [123]. Inspired by this, the Jenner Institute decided to establish their own vaccine platform technology using chimpanzee adenovirus [13]. Vectors utilizing human adenoviruses have the drawback of reduced therapeutic effectiveness owing to the presence of pre-existing neutralizing antibodies against the virus in many individuals [124]. Only a limited number of adults possess neutralizing antibodies against the chimpanzee-derived adenovirus strain Y25 chosen by the Jenner Institute. Therefore, this strain was believed to circumvent pre-existing immunity [75]. ChAdOx1 was designed as a vector vaccine by removing the E1/E3 regions from the Y25 strain, rendering it nonreplicative [75]. With support from the Wellcome Trust, a charitable foundation dedicated to medical research in the UK, the Jenner Institute established a facility to produce adenovirus vector vaccines [13]. Furthermore, they established a facility within the university to manufacture vaccines for clinical trials and a system to create an adenovirus vector vaccine pipeline for testing in clinical trials one after another [13].

Jenner Institute’s researchers Gilbert and Hill, who were leading the research on ChAdOx1, founded a spin-out company named Vaccitech in 2016 with a £10 million investment from Oxford Sciences Innovation [125]. Vaccitech licensed the ChAdOx1 technology and worked on its commercialization efforts [126]. In 2018, Vaccitech secured £20 million in Series A investment from GV, Oxford Sciences Innovation, and Sequoia China [127]. In 2021, they raised $168 million in Series B financing for three early-stage clinical programs targeting chronic hepatitis B virus, human papillomavirus, and prostate cancer, [128]. This Series B financing round was led by M&G Investment Management along with Tencent, Gilead Sciences, the Monaco Constitutional Reserve Fund, Future Planet Capital, and others [128]. The Vaccitech’s technology has attracted significant investments from various investors, supporting research and development efforts.

Supported by these generous investments, ChOxAd1 has been tested in the clinical development of many infectious diseases and cancer vaccines. Before the development of the COVID-19 vaccine, Phase 1–3 clinical trials were conducted for vaccines against influenza, tuberculosis, malaria, meningococcal B, prostate cancer, MERS-CoV, Chikungunya, and Zika [124]. Vaccine development against MERS-CoV was funded by the UK Vaccines Network, which is a partnership between the Department of Health and Social Care and UKRI’s Medical Research Council and Biotechnology and Bioscience Research Council, and this study proved that the ChOxAd1 vaccine was safe and could provoke immune responses [129]. In parallel, the University of Oxford launched the Pandemic X project in 2016, aiming to predict potential future pandemics and engage in responses, treatment development, and infrastructure development [130]. Owing to accumulated efforts in these research and development initiatives, the development of a COVID-19 vaccine using ChOxAd1 progressed rapidly, with the first administration to human subjects commencing 103 days after the genetic sequence of SARS-CoV-2 was publicly disclosed [130].

The key personnel and institutions that contributed to the technological developments in the three cases mentioned above are compiled in Table 2. University-originated scientific and technological seeds have been bridged to university spin-offs by entrepreneurial scientists and other entrepreneurs with investments and support from investors, existing companies, and government agencies, facilitating the practical application of cutting-edge technologies.

### 4.4. The Case of Japan

Japan has the third largest pharmaceutical market in the world after the United States and China [131]. However, Japan’s ability to develop new drugs is weak. For at least the past 30 years, Japan’s pharmaceutical industry has been experiencing a trade deficit that continues to increase [36]. One of the factors behind this is the lack of growth in biotechnology startups, which have significantly contributed to the creation of innovative new drugs in Japan. The importance of university startups has been emphasized in new drug discovery and development as they extensively use scientific knowledge from universities and other academic institutions [132]. Over half of the first-in-class drugs approved by the FDA between 2011 and 2022 were created by small and medium-sized enterprises that were not included in the top 50 pharmaceutical companies in annual revenue [133]. In the U.S., 54% of new drugs approved by the FDA between 2017 and 2022 were originated from relatively new companies established after 1990, whereas in Japan, all new FDA-approved drugs originated from incumbent pharmaceutical companies established before 1980 [134]. This result demonstrates that, unlike the U.S., Japan lacks an ecosystem of biotechnology startups that translate innovative technologies into new drugs. The lack of a drug discovery ecosystem has also posed challenges to the development of COVID-19 vaccines. As of December 2022, no COVID-19 vaccines originated from Japanese companies have been approved worldwide (Table 1). Instead, the Japanese government purchased vaccines developed by foreign companies to vaccinate its citizens, with procurement costs reaching 2.4 trillion yen (approximately $18 billion) by April 2022 [135]. Japan’s pharmaceutical industry trade deficit reached approximately 4.6 trillion yen (approximately $34 billion) in 2022, increasing by over 1 trillion yen (approximately $7.5 billion) compared with 2021, and it is said that this increase is largely attributed to the impact of vaccine imports [36]. The current situation, in which Japan has been unable to create vaccines domestically and has relied on importing foreign-made vaccines, is referred to as “vaccine defeat,” a stark reminder of Japan’s weakness in drug discovery capabilities [136].

In Japan, a major pharmaceutical company has been developing mRNA vaccines since the beginning of the COVID-19 pandemic. However, the company has not published any papers or filed international patents related to mRNA therapeutics, indicating a lack of technological capabilities [12]. Furthermore, before the development of COVID-19 vaccines, there was no track record of developing mRNA therapeutics for other diseases [12]. This lack of technological and clinical capabilities delayed the development of the vaccine, and it was not until 2 August 2023, that the company obtained manufacturing and marketing approval in Japan. However, this vaccine was designed for the initial strain of the virus, and as there is no longer an anticipated demand, it will not be shipped [137]. This company recently applied for approval of this vaccine against mutant strains. However, the Japanese government has been using Pfizer/BioNTech and Moderna mRNA vaccines against the Omicron variant and has announced plans to purchase additional doses of the Omicron XBB variant-specific monovalent vaccine from Pfizer and Moderna [138,139]. Japan continues to rely on COVID-19 vaccines manufactured by other countries, and the situation has not changed.

## 5. Vaccine Development in the Post-COVID-19 Era

Although a vaccine was approved in an extremely short period of approximately one year from the onset of COVID-19, more than 70 million infectious cases and 1.6 million resulting deaths were reported [140]. To accelerate vaccine development and distribution, a strategy was proposed to develop a vaccine within 100 days of the next pandemic [141]. In this regard, the popularization of mRNA and adenoviral vector vaccines has provided a new direction for pandemic vaccine strategies in the post-COVID-19 era. As mentioned, mRNA and viral vector vaccines can be designed rapidly once the viral genome sequence is identified. As seen with COVID-19, viral infections during pandemics can spread rapidly globally, and mutant strains frequently emerge. To address this, it is important to promptly identify the viral genome sequence and take measures, such as vaccine development. In recent years, the development of NGS technology has been instrumental in the rapid identification of genetic sequences of new viruses [142]. In the case of COVID-19, NGS has played a crucial role in the rapid identification of the virus after the outbreak [143]. RNA viruses frequently undergo mutations, making the rapid identification and characterization of mutant strains important. NGS technology has contributed significantly to the detection and identification of new variants of SARS-CoV-2 [144]. Production speed is also an advantage of mRNA and adenoviral vector vaccines. Inactivated vaccines require the cultivation of vaccine strains, which can be time-consuming due to the need for cell cultures or chicken eggs. Moreover, since they require high containment, investments in facility infrastructure are time-consuming and costly. However, the mRNA used in mRNA vaccines is produced *in vitro*, and their scale-up is relatively straightforward, making the manufacturing process simpler than other vaccines [5]. The adenoviruses used in adenoviral vector vaccines are produced by cell culture. The production method is well established, and several improvements have been made to increase the yield and reduce manufacturing cost [145]. Neither technology involves handling actual pathogens, which enhances safety during production. Therefore, mRNA vaccines and viral vector vaccines, which can be designed and manufactured relatively quickly, are well suited for rapid response during pandemic outbreaks and are likely to become standard technology platforms for future pandemic vaccines. Furthermore, it is important to emphasize the exploration of next-generation technologies, such as self-replicating RNA vectors.

For the accelerated approval of the COVID-19 vaccines, the specific regulatory policies also played a crucial role. In the United States, the Operation Warp Speed (OWS) initiative was taken as a national project to accelerate the development and distribution of COVID-19 vaccines [146]. This initiative allowed pharmaceutical companies to utilize data from other vaccines using similar technology platforms and initiate clinical trials in parallel with non-clinical studies [147,148]. The government enhanced the procurement of equipment and materials required for vaccine production and guaranteed the purchase of large quantities of vaccines before the completion of clinical trials [147,148]. These regulatory breakthroughs significantly contributed to the rapid development of the COVID-19 vaccines and could be vaccine development standards in the post-COVID-19 era. The speed of design and production of mRNA and adenoviral vector vaccines align well with new regulatory environments.

However, the emergence of mRNA and viral vector vaccines has not diminished the importance of conventional vaccination technologies. In the case of COVID-19 vaccines, following the global deployment of mRNA vaccines and adenoviral vector vaccines, many inactivated and protein subunit vaccines have been developed (Table 1). Vaccines based on these conventional technologies are necessary even after the practical use of mRNA and viral vector vaccines. This is due to the safety concerns associated with mRNA and viral vector vaccines, which are not observed with traditional vaccines. In the United States, a study investigating the occurrence of myocarditis after administration of Comirnaty or Spikevax found that the risk of myocarditis after receiving mRNA-based COVID-19 vaccines increased across multiple age and sex strata [149]. Data from Denmark showed that vaccination with Spikevax was associated with a significantly increased risk of myocarditis and myopericarditis [150]. In Hong Kong, there is an increased risk of carditis associated with the Comirnaty vaccination, despite no association between CoronaVac and carditis [151]. The increased risk of venous thrombosis and thrombocytopenia is a concern associated with the ChAdOX1 COVID-19 vaccine. A national cohort study in England showed an increased risk of thrombotic episodes and thrombocytopenia within one month of the first dose of the ChAdOx1 vaccine [152]. The risk of thrombosis is supported by a worldwide review of published cases of cerebral venous sinus thrombosis after COVID-19 vaccination [153]. To date, these side effects have not been reported for inactivated vaccines or protein subunit vaccines. Vaccines using conventional technologies have a long record of accomplishment, and knowledge regarding the risks associated with these modalities has accumulated. During periods when a new pathogen is rapidly spreading or when the mortality rate is high, prioritizing speed by rapidly developing and disseminating mRNA or adenoviral vector vaccines is advisable. Subsequently, once the infection rates have subsided and vaccination has become a routine practice, transitioning to vaccines based on conventional technologies with a more accumulated safety profile will likely become common.

The development of the COVID-19 vaccine exemplifies how cutting-edge medical technology saved the world from an unprecedented crisis and served as an opportunity to reevaluate how innovation using advanced science and technology can effectively address societal challenges. Advanced technology development has not occurred overnight, as explained in detail in this study. The practical application of mRNA therapeutics required breakthroughs in various technological aspects and the accumulation of long-term research and development. It took approximately 30 years from the initial concept of mRNA therapeutics at the animal level to the approval of the first mRNA vaccine, while companies such as BioNTech and Moderna took approximately 10 years from their founding to the approval of mRNA therapeutics. Research on ChAdOx1 began in the early 2000s, and it took nearly 20 years to apply to a COVID-19 vaccine. It is important to note that these technologies were not initially developed for COVID-19 vaccines. As detailed in this article, mRNA therapeutics have been explored for many years owing to their potential as cancer vaccines and treatments for rare diseases and others. ChAdOx1 has been tested for the clinical development of various infectious diseases and cancer vaccines. Cutting-edge science and technology often face the challenge of not initially identifying the best applications due to their innovative nature. Technology evolves by exploring various possibilities, and appropriate applications can be discovered through this persistent effort. The accumulation of dedicated research and development over the years is indispensable for the practical application of cutting-edge science and technology. In vaccine development, it is important to continue developing promising foundational technologies invented at universities and other research institutions during peacetime in preparation for a pandemic outbreak. As mentioned earlier, a pharmaceutical company developed a COVID-19 mRNA vaccine in Japan. However, the technological infrastructure was fragile, and rapid vaccine development was not possible. No patents or published papers related to mRNA therapeutics under the name of the pharmaceutical company were identified before the emergence of COVID-19, suggesting that vaccine development was conducted reactively, without sufficient accumulation of technical developments during peacetime. Consequently, mRNA vaccine development lagged significantly behind that of Moderna and BioNTech. By the time the Japanese pharmaceutical company obtained approval, the demand for vaccines against the originally targeted strains had disappeared, leading to a situation in which, despite obtaining approval, the company did not ship the vaccine. The company also applied for the approval of the same vaccine against mutant strains. However, the Japanese government has already imported vaccines for mutant strains from Pfizer and Moderna. As symbolized by this case, it is essential to develop cutting-edge technologies during peacetime to enable swift technological utilization during crises. The presence or absence of such preparedness can sometimes make a significant difference to a nation’s capacity. Emerging technologies may have limited applications and carry risks in their early stages. Nevertheless, policymakers and companies should understand that investing in such early-stage technologies and nurturing them diligently is paramount for future pandemic preparedness. Therefore, it is crucial to allocate appropriate resources for long-term forward thinking and advanced technology development during peacetime.

University startups play an extremely important role in the development of cutting-edge technologies. Many innovative technologies have stemmed from university research. The role of university startups is to explore their applicability and bridge the so-called “valley of death” to practical implementation. In the case of mRNA vaccines, basic research results, such as the discovery of pseudo-uridine by Kariko et al., served as the basis for the establishment of university startups, such as Moderna and BioNTech, leading to the practical application of mRNA therapeutics. ChAdOx1 was developed at the University of Oxford, and Vaccitech, a university startup, played a crucial role in bridging the gap between technology and practical implementation. When university startups make significant progress in research and development, large pharmaceutical companies often provide the substantial funding for clinical trials through licensing or joint development. Pfizer was responsible for the late-stage clinical development of the BioNTech vaccine, whereas AstraZeneca was responsible for the development of the ChAdOx1 vaccine. The research and development efforts for university startups are funded through investments. Moderna received prominent venture capital support at its inception and successfully raised substantial funds from the market. Additionally, they have received significant financial and human resource support from pharmaceutical companies and government agencies. At its inception, BioNTech secured substantial investments from notable investors and received significant market funding. Thus, technological development in university-based ventures would not be viable without a robust investment system to support long-term research and development. The presence of leaders who drive innovation is also crucial. Afeyan, a renowned serial entrepreneur, became the co-founder of Moderna. The husband-and-wife team Şahin and Türeci, who founded BioNTech, were serial entrepreneurs who had previously launched another venture and achieved a successful exit. The development of entrepreneurial talent with a deep understanding of science and technology and a strong commitment to its societal implementation is essential for the practical application of cutting-edge technologies. In other words, an ecosystem that includes university startups that bridge the gap between university technologies and practical implementation, investors and government support for research and development, and entrepreneurial researchers capable of realizing societal implementation must be established and operational. Practical application of advanced technologies cannot be achieved without such a functioning ecosystem. In Japan, which has lagged behind in the development of COVID-19 vaccines, such a drug development ecosystem has not yet been established. University startups are inactive, and the creation of new drugs is still handled by well-established pharmaceutical companies [134]. The venture investment amount in Japan is only 1/100th of that in the United States [35]. Japan’s entrepreneurial activity is significantly lower than the global average [154]. The fragility of the drug development ecosystem is linked to Japan’s weak drug discovery capabilities [36]. The trade deficit in pharmaceuticals has been increasing annually and has further increased due to the import of COVID-19 vaccines [37]. Without the promotion of university startups and their surrounding environments, Japan is likely to lag behind in vaccine development during the next pandemic. The establishment and operation of a startup ecosystem aimed at further advancing vaccine technology is an urgent task for the country.

This article discusses two new vaccine technology platforms, the mRNA and the adenoviral vector vaccines, established through COVID-19 vaccines. The technological details of these platforms are provided, and the direction of future vaccine development strategies is outlined. In addition, this article documents the developmental history of these cutting-edge technologies and emphasizes the importance of an ecosystem comprising universities, investors, governments, and entrepreneurial talent for developing advanced pharmaceutical technologies. This article is expected to provide valuable insights for considering pandemic preparedness in the aftermath of the COVID-19 pandemic.

## Figures and Tables

**Figure 1 vaccines-11-01737-f001:**
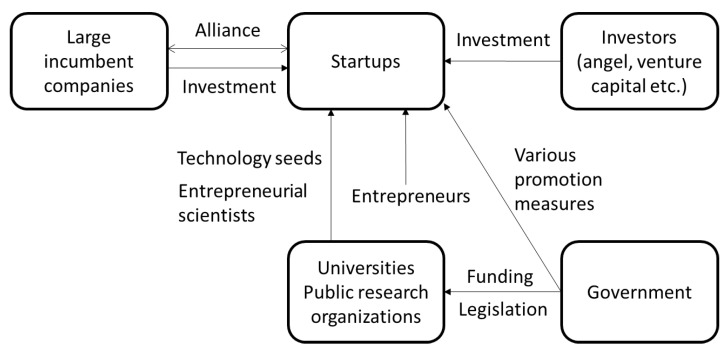
Ecosystem framework in cutting-edge technology development towards commercialization.

**Table 1 vaccines-11-01737-t001:** COVID-19 vaccine ingredients approved as of 2 December 2022.

Vaccine Type	Product Name	Originator
RNA vaccine	GEMCOVAC-19	Gennova Biopharmaceuticals Limited
	Spikevax	Moderna
	Spikevax Bivalent Original/Omicron BA.1	
	Spikevax Bivalent Original/Omicron BA.4/BA.5	
	Comirnaty	BioNTech
	Comirnaty Bivalent Original/Omicron BA.1	
	Comirnaty Bivalent Original/Omicron BA.4/BA.5	
	AWcorna	Walvax
Viral vector vaccine	iNCOVACC	Washington University/Bharat Biotech
	Convidecia (inhaled type of Convidecia: Convidecia Air)	CanSino
	Gam-COVID-Vac (two-dose regimen: Sputnik V one-dose regimen: Sputnik Light)	Gamaleya
	Jcovden	Janssen (Johnson & Johnson)
	Vaxzevria	University of Oxford
DNA vaccine	ZyCoV-D	Zydus Cadila
Inactivated vaccine	Covaxin	Bharat Biotech
	KoviVac	Chumakov Center
	Turkovac	Health Institutes of Turkey
	FAKHRAVAC (MIVAC)	Organization of Defensive Innovation and Research
	QazVac	Research Institute for Biological Safety Problems (RIBSP)
	KCONVAC	Shenzhen Kangtai Biological Products Co
	COVIran Barekat	Shifa Pharmed Industrial Co
	Covilo	Sinopharm
	CoronaVac	Sinovac
	VLA2001	Valneva
Protein subunit vaccine	Zifivax	Anhui Zhifei Longcom/ Chinese Academy of Sciences
	Noora vaccine	Baqiyatallah University of Medical Sciences
	Corbevax	Baylor College of Medicine/Texas Children’s Hospital Center/Dynavax technologies
	Abdala	Center for Genetic Engineering and Biotechnology
	Soberana 02	Finlay Institute
	Soberana Plus	
	V-01	Livzon Mabpharm Inc
	MVC-COV1901	Medigen
	Recombinant SARS-CoV-2 Vaccine (CHO Cell)	National Vaccine and Serum Institute
	Nuvaxovid	Novavax
	IndoVac	PT Bio Farma/Baylor College of Medicine
	Razi Cov Pars	Razi Vaccine and Serum Research Institute
	VidPrevtyn Beta	Sanofi/GSK
	SKYCovione	SK Bioscience/University of Washington
	SpikoGen	Vaxine/CinnaGen Co.
	Aurora-CoV	Vector State Research Center of Virology and Biotechnology
	EpiVacCorona	
Virus-like particles vaccine	Covifenz	Medicago

**Table 2 vaccines-11-01737-t002:** Major resources of ecosystem in mRNA and ChAdOx1 vaccine development.

Type of Resource	Moderna	BioNTech	University of Oxford/Vaccitech
Technology seed/scientific background	-Pseudouridine discovery by Kariko et al. at University of Pennsylvania -Rossi’s research of cell transformation by mRNA at Boston Children’s Hospital	-mRNA cancer vaccine research by Gilboa at Duke University -Cancer immunotherapy research by Şahin and Türeci at University of Mainz	In-house research for novel chimpanzee adenovirus by Gilbert, Hill etc. at Jenner Institute
Entrepreneurial scientist	Derrick J. Rossi	Uğur Şahin, Özlem Türeci	Sarah Gilbert, Adrian Hill
Entrepreneur	Robert S. Langer		
Investor	Noubar Afeyan at Flagship Pioneering	Thomas Strüngmann, Redmile Group	Oxford Sciences Innovation, GV, M&G Investment Management
Incumbent company	AstraZeneca, Alexion, Merck	Pfizer	AstraZeneca
Government agency	DARPA, NIAID		DSC, MRC, BBSRC

## Data Availability

All new data are included in the manuscript.

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
