# Peer review of "mRNA and Adenoviral Vector Vaccine Platforms Utilized in COVID-19 Vaccines: Technologies, Ecosystem, and Future Directions"

_vaccines, 2023, doi:10.3390/vaccines11121737_

Round 1

Reviewer 1 Report

Comments and Suggestions for Authors

Ryo reviewed the new technology platforms used in COVID-19 vaccine development. Not only the technology but also the ecosystem was involved in this manuscript. It’s a interesting topic. Several issues should be addressed prior to acceptance.

Major

1.     The platforms in this manuscript mainly focused on the mRNA and adenovirus. It might be clear to specified the two in the title. Compared to the VLP, the adenovirus could not be considered as a new technology.

2.     For the accelerated approval of the COVID-19 vaccines, the specific regulatory policies also played a crucial role. It might be helpful for the comprehensive understanding of the approval to describe and discuss the regulatory breakthroughs in this process.

3.     When describing the subunit vaccine and VLP vaccines, the author took same examples for the two platform and considered the VLP was superior to subunit in eliciting immune response. How to reach this conclusion?

4.     Line 156-157, “mRNA and viral vector vaccines are similar to protein subunit vaccines in expressing specific antigen proteins within the body”. Subunit vaccines wouldn’t express antigen within the body. Pls rewrite it.

5.     Line 205, undeer should be under.

6.     In line 247-249, an important aspect for low-risk of integration is that the mRNA would not enter the nucleus.

7.     Line 328-336, the main mechanism of immune response induction for live-attenuated or inactivated vaccines is not the DNA or RNA they carry but their structural proteins.

8.     When describing the efficacies of the vaccines, the clinical trial endpoints were different, which made the results not comparable, Pls unify them.

9.     Line 592, tehir should be their.

10.   Line 657, it is recommended to translate yen to US dollars.

Comments on the Quality of English Language

Lots of replicated descriptions were found in this manuscript. Please double-check and carefully review to make the corresponding corrections to make it more concise and readable.

Author Response

Dear anonymous reviewer 1

I sincerely thank you for taking time to review my manuscript and for your valuable comments. Here, I provide a point-by-point response to your review comments.

Comments and Suggestions from reviewer 1

Author’s Reply to the reviewer 1

1.     The platforms in this manuscript mainly focused on the mRNA and adenovirus. It might be clear to specified the two in the title. Compared to the VLP, the adenovirus could not be considered as a new technology.

I changed the title to “mRNA and Adenoviral Vector Vaccine Platforms Utilized in COVID-19 Vaccines: Technologies, Ecosystem, and Future Directions” (in red, line 2 in the new version).

2.     For the accelerated approval of the COVID-19 vaccines, the specific regulatory policies also played a crucial role. It might be helpful for the comprehensive understanding of the approval to describe and discuss the regulatory breakthroughs in this process.

In accordance with the reviewer’s suggestion, I newly created the paragraph to discuss the regulatory breakthroughs that played a crucial role in the accelerated approval of the COVID-19 vaccines in the second paragraph of chapter 5 (in red, line738-749). I have also adjusted reference numbers (in red, line 1165-1170 in the new version).

3.     When describing the subunit vaccine and VLP vaccines, the author took same examples for the two platform and considered the VLP was superior to subunit in eliciting immune response. How to reach this conclusion?

I added more explanation why VLP vaccines induce strong immunity (in red, line 150-151 in the new version). I understood that it is not fair to compare with subunit vaccines therefore deleted the part (line 149 in the new version).

4.     Line 156-157, “mRNA and viral vector vaccines are similar to protein subunit vaccines in expressing specific antigen proteins within the body”. Subunit vaccines wouldn’t express antigen within the body. Pls rewrite it.

To avoid misunderstanding, I rewrote the sentence such as “in that only specific antigen proteins of interest are presented within the body” (in red, line 174 in the new version)

5.     Line 205, undeer should be under.

I corrected (line 223 in the new version).

6.     In line 247-249, an important aspect for low-risk of integration is that the mRNA would not enter the nucleus.

I added the explanation you indicated to the designated part (in red, line 267 in the new version).

7.     Line 328-336, the main mechanism of immune response induction for live-attenuated or inactivated vaccines is not the DNA or RNA they carry but their structural proteins.

In accordance with reviewer’s comment, I rewrote this part (first paragraph of 3.2, in red, line 349-355) with changing the references (in red, line 988-989 in the new version)

8.     When describing the efficacies of the vaccines, the clinical trial endpoints were different, which made the results not comparable, Pls unify them.

The clinical trial endpoints were unified (although the duration after the second dose was different due to limited availability of the information) in Spikevax, Comirnaty, and Vaxzevria (in red, line 341-342, 344-345, 406-408 in the new version). One reference was also changed (in red, line 1014-1015)

9.     Line 592, tehir should be their.

I corrected (line 619 in the current version.

10.   Line 657, it is recommended to translate yen to US dollars.

The data was disclosed only by Yen and translation varies depending on exchange rate. I added approximate US dollars by using average yearly exchange rate between Japanese yen and US dollars in 2022. (in red, line 684-686 in the new version)

Reviewer 2 Report

Comments and Suggestions for Authors

In this review manuscript, the author precisely discussed mRNA and the adenoviral vector vaccine development technologies that highlighted a remarkable success in the rapid development of COVID-19 vaccines for efficiently controlling the COVID-19 pandemic. In addition, the author well described the developmental history of these cutting-edge technologies and emphasized the importance of an ecosystem which enabled the successful development of these advanced technologies. Furthermore, the negative lesson was reasonably discussed based on Japan’s example due to lacking a well-developed ecosystem for the new vaccine technology development. This review provides important insights for the future vaccine development in the post-COVID-19 pandemic. The review is interesting and useful for the future pandemic preparedness. The rationale and objective are clear and fully addressed in the review manuscript.

The following minors may be useful to improve the quality of the review manuscript:

1.    In Introduction, it is better to provide more scientific information to definite each vaccine platform. Based on my understanding, the Inactivated virus, live attenuated virus, viral protein subunit and VLP are the different types of vaccine products which work in different ways to offer protection, while mRNA and viral vector are different types of vaccine technologies or vaccine processing technologies by which how the gene is expressed to formulate a vaccines in vivo. For instance, VLP can be a mRNA-based or viral vector-based VLP vaccine, even an in vitro-produced protein-based VLP vaccine.

2.    Ling 41, “This rapid development” is not clear. For my understanding, the author may intend to state “These rapid vaccine development technologies”.

3.     Lines 114 – 116 “It has also been noted that the inactivation of the pathogens sometimes loses their antigenicity; therefore, adjuvant administration is required to induce strong immunogenicity.” But most vaccination studies including the evaluation of inactivated SARS-CoV-2 virus vaccines indicate that inactivated virus vaccines, which contain the viral genome and are usually made non-infectious by exposure to chemical or physical inactivating agents. Such chemical or physical treatment, such as the most commonly used inactivating agent, formalin, can induce irreversible changes in viral antigens, resulting in poor immunogenicity and weak cell-mediated and mucosal immune responses even with adjuvant administration. Please see following book:

Burrell, C.J.; Howard, C.R.; Murphy, F.A. Chapter 11—Vaccines and Vaccination. In Fenner and White’s Medical Virology, 5th Ed.; Burrell, C.J.; Howard, C.R., Murphy, F.A., Eds.; Academic Press: London, UK, 2017; pp. 155–167.

4.    Line 205, “undeer” is mistyping.

5.    Line 206, “deployment” may be better replaced with “use” for better understanding.

6.    Line 216, “clinically developed foreign-made vaccines” is not clear.

7.    Line 236, “mRNA therapeutic” may be better described as “mRNA therapeutics”.

8.    Line 283-284, “is being utilized” may be better described as “has been utilized”.

9.    Line 592, “tehir” is mistyping.

10. Line 794, “human resources support” may be described as “human resource support”.

11. Self-replicating RNA vector platform is also likely to become standard technology platforms for future pandemic vaccines.

Comments on the Quality of English Language

This review manuscript is well-written, but there are several occurrences such as some typing errors and meaningful confusion that require a proof prior to publication.

Author Response

Dear anonymous reviewer 2

I sincerely thank you for taking time to review my manuscript and for your valuable comments. Here, I provide a point-by-point response to your review comments.

Comments and Suggestions from reviewer 2

Author’s Reply to the reviewer 2

1.    In Introduction, it is better to provide more scientific information to definite each vaccine platform. Based on my understanding, the Inactivated virus, live attenuated virus, viral protein subunit and VLP are the different types of vaccine products which work in different ways to offer protection, while mRNA and viral vector are different types of vaccine technologies or vaccine processing technologies by which how the gene is expressed to formulate a vaccines in vivo. For instance, VLP can be a mRNA-based or viral vector-based VLP vaccine, even an in vitro-produced protein-based VLP vaccine.

In accordance with reviewer’s comments, I added the information to definite each vaccine type in the second paragraph of Introduction (in red, line 33-42 in the new version).

2.    Ling 41, “This rapid development” is not clear. For my understanding, the author may intend to state “These rapid vaccine development technologies”.

I corrected as indicated (in red, line 53-54 in the new version)

3.     Lines 114 – 116 “It has also been noted that the inactivation of the pathogens sometimes loses their antigenicity; therefore, adjuvant administration is required to induce strong immunogenicity.” But most vaccination studies including the evaluation of inactivated SARS-CoV-2 virus vaccines indicate that inactivated virus vaccines, which contain the viral genome and are usually made non-infectious by exposure to chemical or physical inactivating agents. Such chemical or physical treatment, such as the most commonly used inactivating agent, formalin, can induce irreversible changes in viral antigens, resulting in poor immunogenicity and weak cell-mediated and mucosal immune responses even with adjuvant administration.

I added the suggested explanation to the indicated sentence (in red, line 129-132 in the new version)

4.    Line 205, “undeer” is mistyping.

I corrected (line 223 in the new version).

5.    Line 206, “deployment” may be better replaced with “use” for better understanding.

I corrected (line 224 in the new version).

6.    Line 216, “clinically developed foreign-made vaccines” is not clear.

I changed the expression to increase clarity (in red, line 234-235 in the new version)

7.    Line 236, “mRNA therapeutic” may be better described as “mRNA therapeutics”.

I corrected (line 254 in the new version).

8.    Line 283-284, “is being utilized” may be better described as “has been utilized”.

I corrected (line 301-302 in the new version).

9.    Line 592, “tehir” is mistyping.

I corrected (line 619 in the new version).

10. Line 794, “human resources support” may be described as “human resource support”.

I corrected (line 835 in the new version).

11. Self-replicating RNA vector platform is also likely to become standard technology platforms for future pandemic vaccines.

I added this platform at the end of first paragraph of chapter 5 (in red, line 735-737 in the new version).
